# Evaluating the clinical relevance and reliability of outer retinal band length on optical coherence tomography in retinal disease: a cross-sectional study

Rene Cheung ![ORCID],[1,2] Angelica Ly ![ORCID],[1] Henrietta Wang ![ORCID],[2] Michael Kalloniatis ![ORCID],[1,3] Lisa Nivison-Smith ![ORCID][1,2]

This work was presented as a conference abstract at the XXV Biennial Meeting of the International Society for Eye Research, February 19-23, 2023, Gold Coast, Queensland, Australia.

[1]School of Optometry and Vision Science, University of New South Wales, Sydney, New South Wales, Australia
[2]Centre for Eye Health, University of New South Wales, Sydney, New South Wales, Australia
[3]Deakin University, Waurn Ponds, Victoria, Australia

**Correspondence to**
Dr Lisa Nivison-Smith;
l.nivison-smith@unsw.edu.au

## ABSTRACT

**Objectives** Hyper-reflective outer retinal band (HORB) disruptions are reported across a range of retinal disease, yet a reliable, easily implemented assessment method and thorough evaluation of their association to retinal disease is lacking. The purpose of the study was to assess the reliability of using magnitude estimation to evaluate HORB length and determine its association to visual acuity and retinal disease.

**Design** Cross-sectional, retrospective study.

**Setting** Patients attending a secondary eye care clinic in Sydney, Australia.

**Participants** 2039 unique consecutive patients were screened for inclusion between 2 November and 18 January 2021, and 600 were included in the study population. Patients were included if they were referred from primary care, presented for an initial, comprehensive eye examination during the study period, imaged with optical coherence tomography during their visit and over 18 years of age.

**Primary outcome** Reliability of HORB length estimations and associations to clinical outcomes.

**Results** Intragrader (intraclass correlation coefficient, $ICC_{fovea}$=0.81; $ICC_{worst}$=0.91) and intergrader ($ICC_{fovea}$=0.78–0.79; $ICC_{worst}$=0.75–0.88) agreement of HORB length was good to excellent. HORB length was significantly associated with age (p<0.001, β=−0.22 to −0.24) and refractive error (p<0.001, β=0.12–0.16) at all B-scan locations. Visual acuity (p=0.001, β=−0.13) was associated with the primary outcome for foveal B-scans and eccentricity (p=0.002, β=−0.13) and device type (p=0.002, β=0.13) for non-foveal B-scans. Glaucoma was associated with HORB length on univariate analysis (p=0.05–0.06, β=−0.08); however, multivariate analysis revealed no significant association between HORB length and retinal disease.

**Conclusion** HORB length is reliably assessed using magnitude estimation and may be useful as a surrogate biomarker of visual acuity. Several factors affect HORB length estimations, which may contribute to the lack of association to retinal disease and highlights the need for covariable adjustment when examining HORB disruptions.

## STRENGTHS AND LIMITATIONS OF THIS STUDY

⇒ An unselected population was obtained through consecutive recruitment, allowing a robust evaluation of associations between hyper-reflective outer retinal band (HORB) length and retinal disease that is more generalisable to primary eye care settings.

⇒ Magnitude estimation was used to assess outer retinal integrity, which is a reliable psychophysical technique that can be applied to optical coherence tomography imaging without specialised equipment.

⇒ The consecutive recruitment design limited sub-analysis of associations for specific categories of retinal disease.

⇒ Despite previous correlations of external limiting membrane disruptions to retinal disease, it was not included in HORB analysis due to its relatively weak intensity.

⇒ Outer retinal band length was not measured using computerised methods.

## INTRODUCTION

Outer retinal disruptions occur in a number of retinal diseases including retinal dystrophies,[1] pachychoroid spectrum disease,[2] diabetic retinopathy,[3] age-related macular degeneration (AMD)[4] and epiretinal membranes.[5] They are easily observed on optical coherence tomography (OCT) imaging as changes in hyper-reflective, outer retinal band appearance and have been proposed as biomarkers of disease severity,[3 6] recovery following treatment in neovascular AMD,[7] prognosis and visual acuity (VA) recovery in diabetic macula oedema.[8]

Many features have been explored to quantify changes in outer retinal band appearance on OCT with retinal disease including band continuity,[9] reflectivity[10] and thickness.[11] However, few studies have developed methods which emphasise easy translation and access in clinical settings. For example, some groups use custom software to perform

assessments which are not readily available.[11 12] Others evaluate features within a limited area around the foveal centre,[9 13] which would be challenging to view by eye and therefore difficult to incorporate into qualitative evaluations of retinal images routinely performed in clinical practice. Studies also typically evaluate highly selected study populations without control groups,[10 14] increasing the risk of selection bias and impacting the generalisability of findings.

The aim of this study was to assess outer retinal band appearance on OCT using magnitude estimation in an unbiased population. Magnitude estimation is an established, reliable technique for estimating sensory stimuli[15] including line length[16 17] and produces measurements proportional to actual line length on average.[18 19] It has been used to estimate optic nerve cup-to-disc ratio[20] and bulbar conjunctival redness[21] in clinical settings with good reliability. In this study, we determined the reliability of measuring outer retinal band length using this method and assessed its association with clinical outcomes including VA and retinal disease diagnosis in an unselected patient population.

## METHODS
### Study setting
Retrospective patient data were collected from the Centre for Eye Health (CFEH) Sydney, Australia. CFEH is an optometry-led referral centre that provides free diagnostic ocular imaging and disease management services to patients referred by optometrists and medical practitioners for additional assessment and collaborative care services with ophthalmologists. CFEH is a secondary eye care service attended by patients from the community who are suspected of a range of mild to moderate retinal diseases.[22]

### Study population and screening
In total, 2039 unique consecutive patients seen at the CFEH between 2 November 2020 and 18 January 2021 were screened for inclusion in the study. Patients were included in the study if they were: (1) referred from the community; (2) presented for an initial appointment during the study period and (3) were imaged by an OCT device at their visit. Patients were excluded if they were under 18 years of age, did not undergo a complete workup involving diagnosis and management at CFEH, or if the OCT images in their record were below minimum scan quality requirements (Cirrus B-scans: signal strength >6; spectralis scans: quality >15).

### Data extraction
Date of birth, sex, VA, dilation status and diagnosis by CFEH clinicians were subsequently extracted from clinical records and patient reports using practice management software, Bp VIP.net (Best Practice Software). The extracted diagnosis was categorised by two investigators with a combined clinical experience of 20 years into the

following mutually exclusive categories: glaucoma disease, glaucoma suspect, optic nerve disease, macular disease, retinal disease, no ocular pathology, multiple diagnoses, no ocular pathology. All other diagnoses including anterior eye disease and vitreous anomalies were categorised as 'other'. A combined category for outer retinal disease comprised of patients diagnosed with macular or retinal disease was created to further elucidate the association between outer retinal pathology and hyper-reflective outer retinal band (HORB) length. Definitions used for diagnosis classification are described in online supplemental table S1 and patients diagnosed with ageing changes or physiological variations were deemed to be without ocular pathology. The disease eye was selected as the study eye. If both eyes were eligible, the eye with greater disease severity was chosen.

### OCT scan selection
For each patient, two B-scans were extracted from the study eye by an experienced clinician: one through the anatomically identified foveal pit, and the other through the worst area of retinal pathology at a non-foveal location. A non-foveal B-scan was chosen randomly if there was no retinal pathology. B-scans were extracted from the OCT volume with the highest axial resolution which were captured by either the Cirrus HD-OCT (Carl Zeiss Meditec, Dublin, California, USA) or Spectralis OCT (Heidelberg Engineering, Heidelberg, Germany).[23] Details of different scan types extracted for final assessment for all study eyes are described in online supplemental figure S1.

After image extraction, all B-scans were deidentified and randomised to create two separate portable document files of foveal or worst pathology B-scans. The process was then repeated to produce a second set of randomised images for grading to determine the reliability of estimations. Eccentricity of the non-foveal B-scan was calculated from its position relative to the foveal B-scan and interscan distance.

### Magnitude estimation of outer retinal band length
#### HORB assessment
Magnitude estimation involves estimating the magnitude of a stimuli against a reference value and is an established, reliable technique for estimating sensory stimuli[15] including line length.[16–19] This approach was used as the magnitude estimation technique requires subjects to assess relative values (ie, ratios) rather than actual length or size of stimuli in real units. Our approach is also consistent with other applications of magnitude estimation for evaluating ocular structures such as using optic disc size as the reference for cup-disc ratio estimations ratio[20] or the width of a large vein at the disc margin to estimate drusen size in AMD.[24] In this study, the total length of the OCT B-scan was used as the reference and the length of the HORB including the photoreceptor ellipsoid zone (EZ), interdigitation zone with retinal pigment epithelium (IZ) and the retinal pigment epithelium/Bruch's membrane

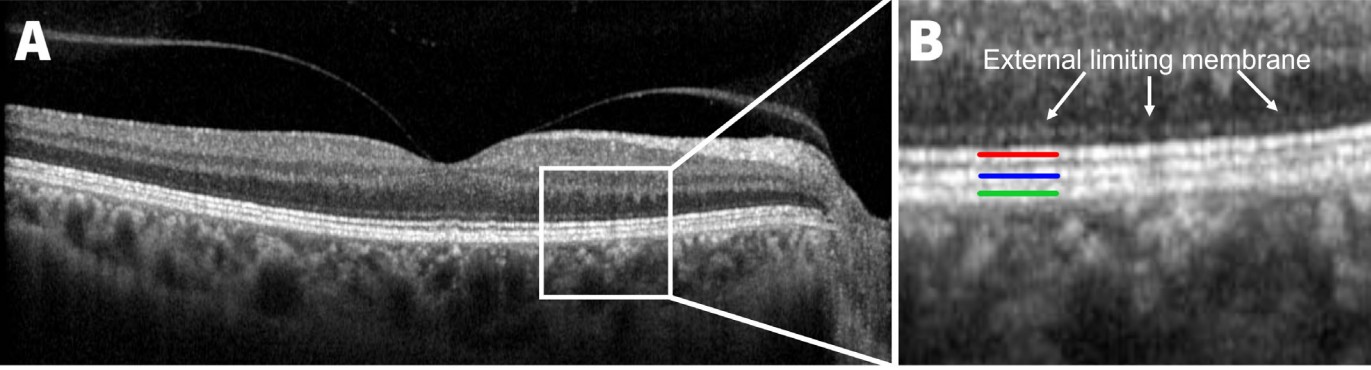

**Figure 1** (A) High definition, Cirrus B-scan of a healthy fovea with (B) the hyper-reflective outer retinal bands (HORB) assessed for outer retinal integrity: ellipsoid zone (red), interdigitation zone (blue) and RPE/Bruch's membrane complex (green). The external limiting membrane (white arrows, B) was not included in our definition of HORB. RPE, retinal pigment epithelium.

complex (RPE)[25] was used as the stimuli (figure 1). Final values were expressed as the estimated percentage of the scan which showed three distinct HORBs (ie, EZ, IZ and RPE) relative to total length. For reference, in the healthy retina, HORB length would be estimated to be near 100% as they are continuous across an OCT B-scan aside from interferences from imaging artefacts, blood vessels or poor image quality and terminate adjacent to the optic disc. An OCT B-scan of a healthy eye with visibility of all three outer retinal bands verified by a senior research clinician of more than 40 years (author MK) was shown to all graders before commencing the task. All graders were clinicians experienced in OCT interpretation to ensure that gradings were not biased by image artefacts such as blood vessel shadowing and only images meeting minimum quality control criteria were included to control for noise.

### Reliability of estimations

Intragrader reliability was determined by assessing the initial and repeat grading performed by one investigator (RC) for all patients (n=600). Repeat gradings were performed 1 week after the initial gradings. Intergrader reliability was determined by comparing independent gradings from RC (grader 1) against HL (grader 2) and AD (grader 3) for half the data set (n=300). Graders 2 and 3 completed all gradings within a 1-month period and both graders were blinded to the image randomisation process and other patient identifiers.

### Statistical analysis

We estimated that a sample size of at least 593 participants would be required to detect a 9% difference in band length between diseased versus non-diseased patients with 85% power to a 90% confidence level. The effect size was determined as the length over which existing grading methods assume that outer retinal bands can be visually discriminated as 'absent' or 'present' (the central 1 mm) over the length of a B-scan (maximum 11 mm). Intra-grader- and intergrader reliability of the magnitude estimation for HORB length was calculated using a two-way mixed-effects model with absolute agreement. A two-way

mixed-effects model was selected as reliability will be assessed using gradings from selected raters rather than randomly drawn raters from a larger population of raters, for which one or two-way random-effects is more appropriate. Intraclass correlation coefficient (ICC) of a single rater was calculated as this reflects typical approaches to estimating clinical features in practice by a single clinician. ICC was interpreted using predefined criteria: <0.5 as poor, 0.5–0.75 as moderate, 0.75–0.9 as good and >0.9 as excellent reliability.[26] The association between HORB length measured by grader 1 and diagnoses was determined using univariable and multivariable regression with p<0.05 considered statistically significant. All statistical analyses were performed using SPSS (V.25; IBM). All summary values are mean±SD unless stated otherwise.

### Patient and public involvement

None.

## RESULTS

### Study population

A total of 2039 patients were screened for inclusion into the study. Patients were excluded if medical consent for research was declined (n=317), they were under 18 years of age (n=37), attended for a follow-up rather than initial appointment (n=1033), did not have OCT performed at visit (n=28) or a complete workup in clinical record (n=4), were referred from tertiary eye care settings (n=12) or did not have a final diagnosis in their record (n=8). The final sample size of 600 participants was sufficient to detect minimum disruptions in band length of 9%. In total, 80.5% of patients (480/600) were diagnosed or identified as at risk of developing ocular disease based on the final diagnosis. The most frequent diagnostic categories were glaucoma suspect (23.8%, 143/600) and macular disease (16.5%, 99/600). No ocular pathology was found in 19.5% of patients (117/600) (online supplemental table S2).

The mean age of all patients was 55.2±15.2 years and 50.3% (302/600) were female. The mean best vision

**Table 1** Patient characteristics (N=600)

| Demographic characteristics | No pathology | Ocular pathology | P value |
| --- | --- | --- | --- |
| Female sex, % (n/N) | 52.1 (61/117) | 49.9 (241/483) | 0.66 |
| Age, mean years (SD) | 49.4 (15.0) | 56.6 (14.7) | **<0.001** |
| Study eye BVS, mean (SD) | −1.21 (2.9) | −0.92 (2.9) | 0.34 |
| Study eye VA (SD) | 0.027 (0.09) | 0.057 (0.13) | **0.005** |
| HORB length estimations | | | |
| Foveal B-scans | 28.9 (28.2) | 26.0 (27.3) | 0.31 |
| Non-foveal B-scans | 22.2 (26.9) | 18.7 (24.2) | 0.17 |
| **OCT scan characteristics** | **No pathology** | **Ocular pathology** | **P value** |
| B-scan length (mm) | 8.2 (1.8) | 10.3 (45.0) | 0.61 |
| Eccentricity (mm) | 154.8 (360.7) | 108.6 (308.9) | 0.16 |
| Device type, % (n/N) (Cirrus OCT) | 82.1 (96/117) | 71.0 (343/483) | **0.02** |

Bolded text represents significant values.
BVS, best vision sphere; OCT, optical coherence tomography; VA, visual acuity.

sphere and VA (logarithm of minimal angle resolution) of the study eye was −0.97 (2.93) dioptres (D) and 0.05 (0.126), respectively (equivalent Snellen VA 6/6.7). Only age (p<0.0001) and VA (p=0.005) were significantly different between patients with and without ocular pathology (table 1).

### HORB length

Examples of different HORB values for eyes with and without pathology are shown in figure 2. Note that these percentages should not be interpreted as absolute measures of band length but rather, indicators of outer retinal integrity that can be easily obtained by eye. The mean (SD) foveal HORB length in patients with and without ocular pathology was 26.0% (27.3) and 28.9% (28.2), respectively; for non-foveal B-scans, 18.7% (24.2) and 22.2% (26.9). There was no significant difference in HORB length for foveal and non-foveal B-scans between groups.

### Reliability of HORB length estimations

Overall, the reliability of HORB length estimations by the same grader and between graders was good to excellent at foveal and non-foveal scan locations according to

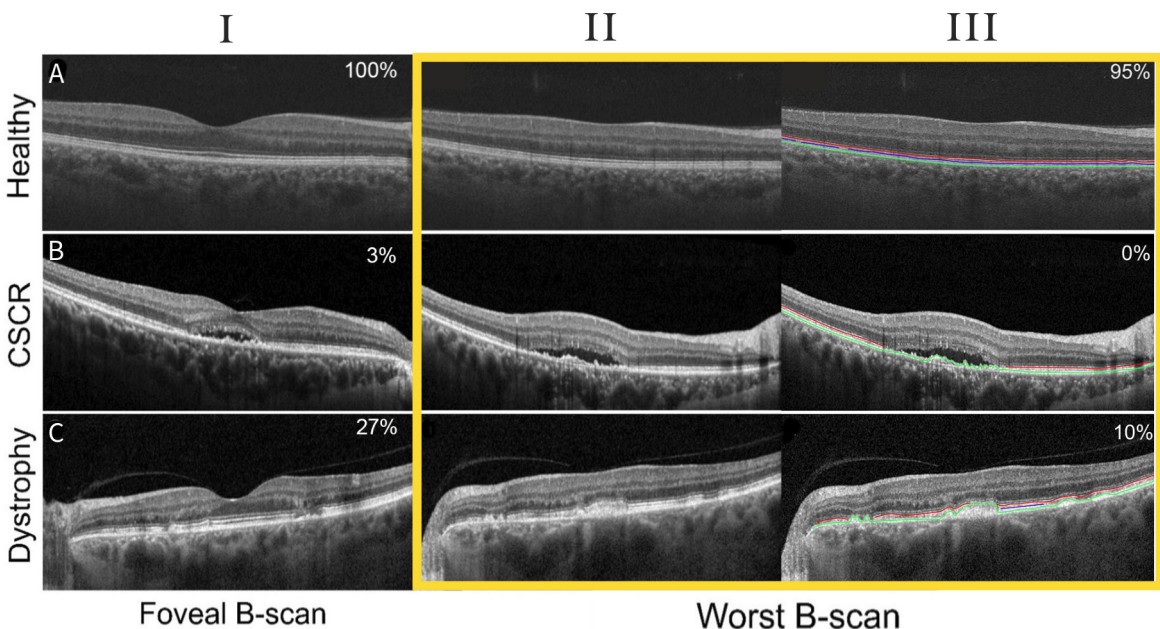

**Figure 2** Examples of foveal (I) and worst B-scans without (II) and with visual representation of outer retinal band estimations performed by eye (III) ellipsoid zone (red), interdigitation zone (blue) and RPE/Bruch's membrane complex (green) for eyes with (row A) no retinal pathology (100%), (row B) central serous chorioretinopathy (0%) and (row C) multifocal pattern dystrophy (10%). Estimation of HORB length is indicated in the top right-hand corner of each panel. RPE, retinal pigment epithelium. HORB, hyper-reflective outer retinal band.

**Table 2** Intraclass correlation results by device type and B-scan length

| Overall | N | | Intra-rater reliability | N | Inter-rater reliability* Grader 2 | Grader 3 |
|---|---|---|---|---|---|---|
| | 600 | Foveal | 0.81 | 298 | 0.79 | 0.78 |
| | | Non-foveal | 0.91 | 296 | 0.75 | 0.88 |
| **Device** | **N** | | **Intra-rater reliability** | **N** | **Inter-rater reliability** | |
| Spectralis | 151 | Foveal | 0.78 | 79 | 0.76 | 0.80 |
| | | Non-foveal | 0.91 | 83 | 0.67 | 0.91 |
| Cirrus | 379 | Foveal | 0.82 | 219 | 0.80 | 0.78 |
| | | Non-foveal | 0.91 | 213 | 0.77 | 0.88 |
| **B-scan length** | **N** | | **Intra-rater reliability** | **N** | **Inter-rater reliability** | |
| 5–7 mm | 123 | Foveal | 0.81 | 94 | 0.79 | 0.81 |
| | | Non-foveal | 0.91 | 33 | 0.75 | 0.90 |
| 7–9 mm | 357 | Foveal | 0.81 | 177 | 0.79 | 0.75 |
| | | Non-foveal | 0.91 | 227 | 0.75 | 0.85 |
| 9–11 mm | 34 | Foveal | 0.57 | 13 | 0.80 | 0.96 |
| | | Non-foveal | 0.94 | 21 | 0.79 | 0.91 |
| 11–15 mm† | 2 | Foveal | 0.90 | 2 | – | 0.83 |
| | | Non-foveal | 0.88 | 2 | – | 0.88 |
| 15–17 mm | 17 | Foveal | 0.76 | 12 | 0.67 | 0.87 |
| | | Non-foveal | 0.95 | 13 | 0.70 | 0.83 |

*Inter-rater reliability was calculated from approximately 50% of foveal and non-foveal images from non-identical subsamples of participants.
†Insufficient values to calculate agreement as both graders assigned a value of 0–1 of the cases in this group.

predefined criteria (ICC=0.75–0.91).[26] A subgroup analysis of reliability by OCT device and B-scan length also found that the intra and inter-rater reliability of HORB estimations for B-scans from Cirrus OCT (ICC=0.77–0.91) and B-scans 5–9 mm in length (ICC=0.75–0.91) was good to excellent. For longer B-scans and those captured by Spectralis OCT, the intra and intergrader reliability was at least moderate (ICC=0.57–0.96) (table 2). Note that the subset of patients with inter-rater reliability data is slightly less than 300 due to processing errors.

### Association between HORB length and retinal disease

For foveal B-scans, age (p<0.001), VA (p<0.001), BVS (p=0.006) and a diagnosis of glaucoma (p=*0.03*) were significantly associated with HORB length on univariate analysis. A diagnosis of macular disease or outer retinal disease showed near significant associations (p=0.06, p=0.06, respectively). When all variables which demonstrated a p<0.25 on univariable analysis were included into multivariable analysis, only age, VA and BVS remained significant (table 3).

For non-foveal B-scans, age (p<0.001), VA (p=0.001), device type (p=0.01) and glaucoma diagnosis (p=*0.03*) were significantly associated with HORB length on univariable analysis. A diagnosis of macular disease again showed a near significant association (p=0.06). When all variables demonstrating p<0.25 in univariable analysis were included in multivariable analysis, only age, BVS,

eccentricity and device type were significantly associated with HORB length (table 3).

Together, these results showed no significant association between HORB length and retinal disease. However, age and refractive error significantly affected HORB length at both the foveal and non-foveal locations. VA had a significant effect on foveal HORB length, while B-scan length and eccentricity significantly affected the HORB length of non-foveal B-scans. A subanalysis of patients without pathology was also conducted to assess for disease-related confounding factors (online supplemental table S3). Associations were preserved for age for foveal and non-foveal B-scans. For foveal B-scans, associations to VA (p=0.09) and refractive error (p=0.089) were no longer significant, as for eccentricity (p=0.93) and device type (p=0.18) for non-foveal B-scans in eyes without ocular disease, contrasting with the whole cohort analysis.

### DISCUSSION

This study found that HORB length could be assessed reliably from OCT B-scans using magnitude estimation. HORB length estimations were significantly associated with age and VA; however, they were not significantly associated with retinal disease after adjusting for covariables. This points to the importance of assessing the utility of

**Table 3** Association between patient/scan characteristics and HORB length

| Univariable analysis | Foveal B-scans | | Non-foveal B-scans | |
|---|---|---|---|---|
| | P value | β* | P value | β |
| Age (years) | **<0.001** | −0.21 | **<0.001** | −0.33 |
| Sex (reference: female) | 0.64 | −0.02 | 0.87 | 0.33 |
| VA (logMAR) | **<0.001** | −0.20 | **0.001** | −27.06 |
| BVS (dioptres) | **0.006** | 0.11 | 0.14 | 0.51 |
| B-scan length (mm) | 0.08 | 0.07 | 0.31 | −0.04 |
| Eccentricity (mm from fovea) | – | – | 0.13 | −0.06 |
| Device type† (reference: spectralis OCT) | 0.40 | 0.04 | **0.01** | 0.11 |
| Diagnosis (reference: absence) | | | | |
| Glaucoma disease | **0.03** | −0.08 | 0.03 | −0.08 |
| Glaucoma suspect | 0.81 | 0.83 | 0.38 | −2.71 |
| Macular disease | 0.06 | −7.00 | 0.06 | −6.25 |
| Retinal disease | 0.40 | −3.45 | 0.61 | −1.88 |
| Optic nerve disease | 0.83 | 1.27 | 0.97 | −0.23 |
| Outer retinal disease | 0.06 | −4.61 | 0.24 | −2.63 |
| **Multiple linear regression analysis‡** | | | | |
| Age (years) | **<0.0001** | −0.22 | **<0.0001** | −0.24 |
| VA (logMAR) | **0.001** | −0.13 | 0.09 | −0.07 |
| BVS (dioptres) | **<0.001** | 0.16 | **<0.001** | 0.12 |
| B-scan length (mm) | 0.07 | 0.07 | – | – |
| Eccentricity (mm from fovea) | – | – | **0.002** | −0.13 |
| Device type (reference: spectralis OCT) | – | – | **0.002** | 0.13 |
| Diagnosis (reference: absence) | | | | |
| Glaucoma disease | 0.24 | −0.05 | 0.25 | −0.05 |
| Macular disease | 0.93 | 0.005 | 0.99 | −0.001 |
| Outer retinal disease | 0.29 | −0.06 | 0.87 | −0.01 |

*Standardised.
†Spectralis or Cirrus device.
‡Includes all univariable analysis variables demonstrating p≤0.25.[43]
BVS, best vision sphere; HORB, hyper-reflective outer retinal band; logMAR, logarithm of minimal angle resolution; OCT, optical coherence tomography; VA, visual acuity.

outer retinal biomarkers in large, unselected patient populations and controlling for all confounders.

### Reliability

HORB length can be reliably assessed using magnitude estimation across different OCT devices and scan protocols in patients with and without retinal pathology. Our results show that the reliability of estimating HORB length at the fovea was good to excellent across two major commercially available OCT devices and scan lengths of 9 mm or less for an unselected population exhibiting a range of ocular pathology. Moreover, intrarater and inter-rater reliability of HORB estimations was at least moderate for all devices or scan lengths tested at both foveal and non-foveal B-scan locations. At non-foveal locations, intrarater reliability of magnitude estimations was excellent for both OCT devices and all scan lengths and

only moderate to good between graders, with the exception of B-scans captured by Spectralis OCT and B-scan lengths of 7 mm or less. The greater axial resolution of Spectralis OCT B-scans (vs Cirrus OCT[23]) and increased difficulty of estimating HORB length across longer B-scan likely accounts for these findings.

Our results are in line with the few previous reports that found external limiting membrane (ELM) and EZ integrity measurements reliable.[7 27] Indeed, other applications of magnitude estimation in eye care such as optic nerve cup-to-disc ratio estimation[20] and bulbar conjunctival redness[21] grading, also show good reliability and are routinely used in clinical practice. While categorical approaches have been used previously to evaluate outer retinal integrity, many do not report reliability and the continuous nature of magnitude estimations may be

advantageous over these methods as visual assessment of clinical features using scales with smaller increments has been shown to improve inter and intrarater reliability.[28 29]

### Association between outer retinal disruptions and clinical outcomes

We found VA was significantly associated with foveal HORB length in an unselected patient population. This supports previous suggestions that outer retinal integrity could be a useful surrogate biomarker of VA,[30] particularly in severe retinal disease such as retinitis pigmentosa where VA measures become less repeatable or unobtainable.[31] HORB length was also significantly associated with age in both foveal and non-foveal scans, consistent with other reports,[10 12] emphasising the need for age correction when assessing image-based biomarkers.

We also identified refractive error, eccentricity and device type as additional covariables affecting HORB length that have not been previously reported. However, in our subset analysis of eyes without ocular disease, only age remained significantly associated with both foveal and non-foveal HORB length while refractive error significantly affected non-foveal HORB but not foveal HORB length. These differences could be explained by the effects of cataracts and pseudophakia on VA and refraction, particularly as the diseased group was statistically older. The lack of association between B-scan eccentricity and device type to HORB length in eyes without ocular pathology likely stems from differences in clinical decision-making processes between groups as non-foveal B-scans are selected randomly in eyes without pathology and OCT devices may be chosen based on imaging capabilities. For example, peripheral OCT B-scans are more easily obtained using Spectralis OCT devices[32] and thus are more likely to be used when peripheral pathology is identified.

Differences in cone density with axial length[33] may explain the association between refractive error and HORB length as band reflectivity varies with cone density[34] and affects band visibility. Lower cone densities are also found at greater eccentricities which may explain the negative correlation between HORB length and eccentricity in non-foveal scans. This effect was only significant for non-foveal scans, perhaps as foveal scans were arbitrarily selected and did not necessarily include outer retinal disruptions. Considering the number of studies emphasising the prognostic potential of outer retinal disruptions[7 8] and relationships to other clinical outcomes, the significant covariables identified in this study will be highly relevant to future validation of these biomarkers in retinal disease.

Other groups have reported significant reduction of EZ intensity and IZ area in glaucomatous eyes compared with age-matched normal eyes.[35 36] We also found that glaucoma disease was significantly associated with HORB length but this was not maintained when we adjusted for covariables in multivariate analysis. Similarly, outer retinal disease such as subretinal fluid[11] and early AMD[4]

have been associated with compromised HORB integrity, yet we did not find a significant association after adjusting for covariables. The disparity between our study findings and previous evidence may stem from the recruitment of study participants from a non-tertiary care centre rather than hospital or ophthalmology clinic settings[2 5 7 11] where retinal disease patients are typically at more advanced stages and exhibit a higher degree of outer retinal disruption.[3 5] Thus, examining magnitude estimation of HORB length in a population with more severe retinal disease may be useful to elucidate the strength of its association to disease.

Mean HORB length estimations were also lower than expected, particularly in the group without ocular pathology where we expected to see intact HORB at least throughout the foveal scan. The strong inverse correlation between age, VA, B-scan length and eccentricity and HORB length suggests that these factors likely contributed to this result as the effect was sustained after adjusting for disease status. These findings again emphasise the importance of adjusting for covariables when measuring outer retinal band biomarkers.

### Strengths and limitations

A key strength to the study was using a consecutive recruitment strategy to identify an unselected patient population, which was more representative of individuals attending primary eye care practices. This, therefore, improves the generalisability of findings by enabling a true estimation of associations between HORB length, clinical outcomes and covariables. However, this also created limitations around subgroup sample sizes due to the low prevalence of retinal disease in patients referred to CFEH.[22] As such, further work is needed to evaluate outer retinal disruptions in specific disease types and may benefit from oversampling target subgroups.

Compared with previous groups, we used a reliable psychophysical technique to assess outer retinal integrity by eye which uses a continuous rather than categorical scale (eg, absent/discontinuous vs present/continuous)[37 38] This is advantageous as visual assessment of clinical features using scales with smaller increments has been shown to improve inter-rater and intrarater reliability.[28 29] A limitation of the study was that the ELM was not included as part of the HORB assessment even though ELM disruptions have also been correlated with retinal disease.[3 13 14] This structure was not included in the grading task as the intensity of the ELM is markedly reduced compared with the other bands and may be difficult to reliably detect by visual estimation. Given our findings, it may be worthwhile evaluating the impact of patient and imaging factors on ELM integrity in future investigations.

Another limitation was that all estimations were performed manually and no artificial (AI) intelligence-driven methods of quantifying HORB length were explored. While AI techniques have shown increasing promise for clinical applications,[39] the implementation

of AI-driven tools in primary or tertiary eye care clinics is presently low with only 15.7% of ophthalmologists currently using AI-derived algorithms in daily practice.[40] Primary eye care practices are also known to have slow uptake of new clinical tools[41] with clinicians still reporting reservations about AI decision-support technology.[42] Thus, magnitude estimation was pursued as a highly accessible technique that requires no specialised equipment to initially assess whether subjective outer retinal integrity gradings are indeed associated with retinal disease, as described by other groups, and whether the method can be used to establish a reliable ground truth. Our results suggest that the clinical relevance of outer retinal integrity assessments to retinal disease requires further investigation using a larger pool of raters and greater representation of disease subgroups in which associations neared significance. If significant associations to disease are confirmed, then magnitude estimations should be validated against computationally derived measurements of outer retinal band length before potential deployment to clinical settings.

## CONCLUSION

We show that outer retinal band length can be reliably assessed using magnitude estimation and may be useful as a surrogate biomarker of VA. Importantly, this study is the first to assess the reliability of measuring HORB integrity beyond the fovea and demonstrates that HORB length at non-foveal locations can be measured reliably using magnitude estimation. Several factors including age, refractive error, eccentricity and device type affected HORB length estimations and highlights the importance of adjusting for covariables when evaluating HORB disruptions in retinal disease to ensure true associations are obtained.

**Acknowledgements** The authors thank Natalie Eshow, Meri Galoyan, Helene Ly and Matt Trinh for their assistance on data collection and statistical analysis.

**Contributors** RC, LN-S, AL and MK were involved in study conception and design. HW and RC performed data collection and RC analysed and drafted the manuscript. RC, LN-S, AL and MK participated in the interpretation of results and revision of the manuscript. The corresponding author confirms that all listed authors meet authorship criteria and no other meeting criteria have been omitted. LN-S is responsible for the overall work as the guarantor.

**Funding** This work was supported by funding from the National Health and Medical Research Council (NHMRC #1174385, #1186915) and the Research Training Program scholarship. Guide Dogs NSW/ACT provides support for the Centre for Eye Health (the clinic from which patient data were obtained).

**Competing interests** None declared.

**Patient and public involvement** Patients and/or the public were not involved in the design, or conduct, or reporting, or dissemination plans of this research.

**Patient consent for publication** Not applicable.

**Ethics approval** This retrospective, cross-sectional study was approved by the Biomedical Human Research Ethics Advisory Panel of the University of New South Wales (Sydney, Australia) (HC190746; June 2020). Written patient consent for use of medical data for research is obtained at the initial consultation at the Centre for Eye Health as standard protocol in line with the Declaration of Helsinki, which is made available for retrospective analysis.

**Provenance and peer review** Not commissioned; externally peer reviewed.

**Data availability statement** Data are available on reasonable request. Further information about procedures for obtaining and accessing data from Centre for Eye Health is described at https://www.centreforeyehealth.com.au/research/ (email: enquiries@cfeh.com.au).

**ORCID iDs**
Rene Cheung http://orcid.org/0000-0003-4422-9037
Angelica Ly http://orcid.org/0000-0001-7881-1522
Henrietta Wang http://orcid.org/0000-0002-6694-7622
Michael Kalloniatis http://orcid.org/0000-0002-5264-4639
Lisa Nivison-Smith http://orcid.org/0000-0001-6677-1949

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
