## [Reviewer comments · BMJ Open]

ARTICLE DETAILS

TITLE (PROVISIONAL)	Evaluating the clinical relevance and reliability of outer retinal band length on optical coherence tomography in retinal disease: a cross-sectional study
AUTHORS	Cheung, Rene; Ly, Angelica; Wang, Henrietta; Kalloniatis, Michael; Nivison-Smith, Lisa

VERSION 1 – REVIEW

REVIEWER	Ying, Gui-shuang University of Pennsylvania Perelman School of Medicine, Ophthalmology
REVIEW RETURNED	13-Sep-2023

GENERAL COMMENTS	The study evaluated the reliability and clinical relevance of outer retinal band length on OCT. While the paper provided some useful information, the paper can be improved as commented below. 1. What is the time interval for the initial grading and regrading for the outer retinal band disruptions? Were all OCT images had initial grading and regrading of outer retinal band disruptions by two different graders?2. In the evaluation of factors associated outer retinal band length, what grading results were used for analysis? Were data from initial grading and regrading from both graders all included into analysis, or average of two graders from initial grading used for analysis? Please clarify and make it clear in statistical analysis section.3. It will be good to use the Bland-Altman plot for demonstrating the inter-grader and intra-grader agreement for outer retinal band length.4. Please clarify how two-way random mixed-effects model was used to evaluate inter-grader and intra-grader agreement. What are the two factors for the two-way random mixed effect model? How the intraclass correlation was calculated from the mixed effect model?5. In Table 2, why the N by device and by B-scan length did not add to 600 eyes? For B-scan length, why there is no category of 11-15 mm?6. Please provide the number of eyes in each category of the ocular diagnosis.7. In Table 3, for the row corresponding to diagnosis, how can there is one beta-coefficient? Since diagnosis is a categorical variable that include several ocular disease categories, there should be several beta-coefficients. Please provide the mean outer retinal band length for each ocular diagnosis.8. Table 3, please clarify what beta-coefficient corresponds to. For example, for beta-coefficient for sex, what is the reference group, for beta-coefficient of age, does this coefficient correspond to per
---

	year increase in age? For beta coefficient of diagnostic category, what is the reference group? 9. The relationship of outer retinal band length with age, gender, visual acuity, refraction and eccentricity should be evaluated among eyes without any ocular disease. Current evaluation among all subjects with mixture of various ocular disease is complicated, and there may be many other confounding factors associated disease can distort the association.
--	---

REVIEWER	Yao, Xincheng University of Illinois at Chicago
REVIEW RETURNED	01-Oct-2023

GENERAL COMMENTS	This manuscript examined the outer retinal band appearance observed in OCT images using magnitude estimation. The subject involved for this study included an unbiased population. The reliability of measuring outer retinal band length using this method has been determined and its association with clinical outcomes has been assessed. The relevance of using such measurement for the examination of visual acuity and retinal disease diagnosis in an unselected patient population has been studied. This study concluded that the hyper outer reflective retinal band (HORB) length is reliably assessed using magnitude estimation and may be useful as a surrogate biomarker of visual acuity. Several factors affect HORB length estimations, which may contribute to the lack of association to retinal disease and highlights the need for covariable adjustment when examining HORB disruptions. The rationale behind magnitude estimation as a reliable technique in OCT B-scans has not been established clearly. The presented results were obtained manually by only two graders. A thorough comparative assessment has not been conducted to validate the reliability of this approach. The method section needs more clarity with better visual depiction. Considering these issues, further improvements are required for publication. Specific comments:  1. The rationale behind using length of OCT B-scan as the reference and the length of the hyperreflective outer retinal bands (HORB) as the stimuli has not been discussed in detail. Other than pathological conditions, band disruptions in OCT can also occur due to noise artefacts, aberration, and even blood vessels. Hence the authors need to explain if this method is a well-established approach in OCT. 2. The method section is not documented intricately enough. The term "magnitude estimation" needs to be explained comprehensively with visual representation or it may create confusion amongst the reader as the term magnitude is often used to refer to the signal strength in OCT recording. 3. In Figure 2, what is the basis of rating the B-scans in different panels with different percentage values? Does it follow any rigorous method defined by numerical parameters? 4. All the estimations were done manually which are subject to human error. Hence the results need to be compared with established computational methods to verify the reliability of the human estimation. It is true that AI tools are not widely used in primary or tertiary eye clinics. However, the computational methods can provide a baseline for comparing the estimated HORB values that were measured manually. 5. The authors need to include more graders in this study for a better statistical analysis and validation. It looks not convincible to
---

	discuss intergrader agreement with only two graders involved in this study.
--	---

VERSION 1 – AUTHOR RESPONSE

Responses to Reviewer 1: Dr. Gui-shuang Ying, University of Pennsylvania Perelman School of Medicine, University of Pennsylvania Perelman School of Medicine

Comment 10: What is the time interval for the initial grading and regrading for the outer retinal band disruptions? Were all OCT images had initial grading and regrading of outer retinal band disruptions by two different graders?

Response: More detail on grading processes has been provided under the ‘reliability of estimations’ section in the methods:

Intra-rater reliability was determined by assessing initial and repeat gradings performed by one investigator (RC) for all patients (n=600). Repeat gradings were performed one week after the initial gradings. Inter-grader reliability was determined by comparing independent gradings from RC (grader 1) against HL (grader 2) and AD (grader 3) for half the data set (n =300). Grader 2 and 3 completed all gradings within a one-month period and both graders were blinded to the image randomization process and patient identifiers. (Page 5, lines 183-189)

Comment 11: In the evaluation of factors associated outer retinal band length, what grading results were used for analysis? Were data from initial grading and regrading from both graders all included into analysis, or average of two graders from initial grading used for analysis? Please clarify and make it clear in statistical analysis section.

Response: The gradings used for univariable and multivariable regression were from the initial gradings of grader 1. This was done because the agreement between graders was good to excellent (ICC ≥ 0.75) and including estimations for 100% rather than 50% of the participant cohort provides a more powerful assessment of associations. The statistical analysis section has been modified to clarify this:

The association between HORB length *measured by grader 1* and diagnoses was determined... (Page 6, line 206)

Comment 12: It will be good to use the Bland-Altman plot for demonstrating the inter-grader and intra-grader agreement for outer retinal band length.

Response: Thank you for the suggestion to improve our analysis of results. The authors agree that a Bland-Altman plot is helpful for assessing reliability. However, this has not been included as the mean differences in HORB length gradings obtained within and between graders for foveal and non-foveal B-scans do not follow a normal distribution (Kolmogorov-Smirnov test, $p < 0.0001$), which is an assumption for conducting the Bland-Altman test. Using logarithmic transformed data of estimation differences between graders also resulted in non-normally distributed results ($p < 0.05$).¹³ We have however strengthened our assessment of inter-rater reliability by including an additional grader (see comment 19 and 24).

Comment 13: Please clarify how two-way random mixed-effects model was used to evaluate inter-grader and intra-grader agreement. What are the two factors for the two-way random mixed effect model? How the intraclass correlation was calculated from the mixed effect model?

Response: Intraclass correlation calculations are based on a practical guide developed by Koo et al.¹⁴ for choosing the correct form of reliability analyses in clinical research. Specifically, we chose a two-way mixed-effects model with absolute agreement to assess intra- and inter-grader reliability as the direction of effect was unknown (hence two-way model) and the raters used were of a selected

population rather than randomly drawn from a larger population (hence mixed-effects vs random-effects model) We have clarified this in the methods as detailed below:

Intra- and inter-grader reliability of the magnitude estimation for HORB length was calculated using a two-way mixed-effects model with absolute agreement. A two-way model mixed-effects model was selected as reliability will be assessed using gradings from selected raters rather than randomly drawn from a larger population of raters, for which one or two-way random-effects is more appropriate. Intraclass correlation (ICC) of a single rater was calculated as this reflects typical approaches to estimating clinical features in practice by a single clinician. (Page 5-6, lines 199-204)

Comment 14: In Table 2, why the N by device and by B-scan length did not add to 600 eyes? For B-scan length, why there is no category of 11-15 mm?

Response: Thank you for identifying this inconsistency. The total sample size for B-scan length categories has been updated for foveal and non-foveal B-scans after it was found that some values were inadvertently omitted in the original analysis. The inter-rater reliability results have been re-analysed to account for these changes, resulting in more conservative estimates of reliability for foveal B-scans and improved inter-rater reliability for non-foveal B-scans. Following these adjustments, only three minor results changed in significance:

Reliability

- The inter-rater reliability of non-foveal B-scans obtained from Spectralis devices is revised down from 'good' (0.79) to 'moderate' (0.67) and improved from 'moderate' (0.72) to 'good' (0.77) for non-foveal B-scans obtained from Cirrus OCT devices. The results still show overall good reliability across the two device types.
- The inter-rater reliability of HORB estimations for foveal B-scans is revised down from 'excellent' (0.91) to 'good' (0.79) for B-scans 5-7mm in length and 'good' (0.80) to 'moderate' (0.67) for B-scans 15-17mm in length. Conversely, inter-rater reliability improved from 'moderate' (0.54) to 'good' (0.79) for non-foveal B-scans 9-11mm in length, resulting in more consistent estimates of inter-rater reliability across different B-scan locations.

Note that whilst the ICC values were altered, the overall conclusions of this section were unchanged after updating sample size. The 11-15mm category has been added to Table 2 in the revised manuscript. In addition, a footnote has been added to Table 2 (changes highlighted in green) to highlight that inter-rater reliability was assessed for a subset of patients. Note that the final sample size of participants with inter-rater reliability data is slightly less than 300 from processing errors. (Page 7, lines 252-253).

Table 2: Intraclass correlation results by device type and B-scan length

Overall	N	Intra-rater reliability	N	Inter-rater reliability*	
				Grader 2	Grader 3
	600	Foveal	298	0.79	0.78
		Non-foveal	296	0.75	0.88
Device	N	Intra-rater reliability	N	Inter-rater reliability	
Spectralis	151	Foveal	79	0.76	0.80
		Non-foveal	83	0.67	0.91
Cirrus	379	Foveal	219	0.80	0.78
		Non-foveal	213	0.77	0.88
B-scan length	N	Intra-rater reliability	N	Inter-rater reliability	

5-7mm	123	Foveal	0.81	94	0.79	0.81
		Non-foveal	0.91	33	0.75	0.90
7-9mm	357	Foveal	0.81	177	0.79	0.75
		Non-foveal	0.91	227	0.75	0.85
9-11mm	34	Foveal	0.57	13	0.80	0.96
		Non-foveal	0.94	21	0.79	0.91
11-15mm†	2	Foveal	0.90	2	-	0.83
		Non-foveal	0.88	2	-	0.88
15-17mm	17	Foveal	0.76	12	0.67	0.87
		Non-foveal	0.95	13	0.70	0.83

*Inter-rater reliability was calculated from approximately 50% of foveal and non-foveal images from non-identical sub-samples of participants. †Insufficient values to calculate agreement as both graders assigned a value of zero to one of the cases in this group.

The results have been also updated to reflect these changes:

A sub-group analysis of reliability by OCT device and B-scan length also found that the intra and inter-rater reliability of HORB estimations for B-scans from Cirrus OCT (ICC=0.75-0.91) and B-scans 5-9mm in length (ICC=0.75-0.91) was good to excellent. For longer B-scans and those captured by Spectralis OCT, the intra and inter-grader reliability was at least moderate (ICC=0.57-0.96) (Table 2). (Page 7, lines 248-252).

Univariable and multivariable regression were re-performed, resulting in one minor change:

Regression analyses

- B-scan length is not a significant factor affecting HORB estimations at foveal and non-foveal locations on univariate or multivariate regression analysis whereas OCT device type is significantly associated with non-foveal B-scan HORB estimations after re-analysis. This may reflect differences in peripheral imaging capacity between the two OCT devices examined, which is addressed in the discussion

The lack of association between B-scan eccentricity and device type to HORB length in eyes without ocular pathology likely stems from differences in clinical decision-making processes between groups as non-foveal B-scans are selected randomly in eyes without pathology and OCT devices may be chosen based on imaging capabilities. For example, peripheral OCT B-scans are more easily obtained using Spectralis OCT devices¹⁵ and thus are more likely to be used when peripheral pathology is identified. (Page 10, lines 330-335)

Changes highlighted in green:

Table 3: Association between patient/scan characteristics and HORB length

Univariable analysis	Foveal B-scans		Non-foveal B-scans	
	p	β^{\dagger}	p	β
Age (years)	<0.001	-0.21	<0.001	-0.33
Sex (reference: female)	0.64	-0.02	0.87	0.33
VA (logMAR)	<0.001	-0.20	0.001	-27.06
BVS (Dioptres)	0.006	0.11	0.14	0.51
B-scan length (mm)	0.08	0.07	0.31	-0.04
Eccentricity (mm from fovea)	-	-	0.13	-0.06
Device type‡ (reference: Spectralis OCT)	0.40	0.04	0.01	0.11
Diagnosis (reference: absence)				
Glaucoma disease	0.03	-0.08	0.03	-0.08
Glaucoma suspect	0.81	0.83	0.38	-2.71
Macular disease	0.06	-7.00	0.06	-6.25
Retinal disease	0.40	-3.45	0.61	-1.88
Optic nerve disease	0.83	1.27	0.97	-0.23

Outer retinal disease	0.06	-4.61	0.24	-2.63
Multiple linear regression analysis[§]				
Age (years)	<0.0001	-0.22	<0.0001	-0.24
VA (logMAR)	0.001	-0.13	0.09	-0.07
BVS (Dioptres)	<0.001	0.16	<0.001	0.12
B-scan length (mm)	0.07	0.07	-	-
Eccentricity (mm from fovea)	-	-	0.002	-0.13
Device type (reference: Spectralis OCT)	-	-	0.002	0.13
Diagnosis (reference: absence)				
Glaucoma disease	0.24	-0.05	0.25	-0.05
Macular disease	0.93	0.005	0.99	-0.001
Outer retinal disease	0.29	-0.06	0.87	-0.01

BVS = best vision sphere; VA = visual acuity; † Standardised; ‡ Spectralis or Cirrus device; § Includes all univariable analysis variables demonstrating $p \leq 0.25$.¹⁶

Comment 15: Please provide the number of eyes in each category of the ocular diagnosis.

Response: Details on the number of eyes has been added to Table S2 of the Supplementary Material as suggested (page 2, lines 12-13):

Diagnostic category	(%, n/N)	Foveal HORB length (mean, SD)	Non-foveal HORB length (mean, SD)
Glaucoma suspect	23.8 (143/600)	27.9 (27.4)	18.2 (24.8)
Macular disease	16.5 (99/600)	21.9 (26.4)	16.0 (21.8)
Retinal disease	12.0 (72/600)	25.5 (29.4)	20.3 (26.7)
Multiple diagnoses	10.8 (65/600)	21.8 (24.6)	16.8 (20.3)
Other diagnoses	9.3 (56/600)	31.1 (29.7)	23.5 (28.0)
Optic nerve disease	4.5 (27/600)	30.2 (25.3)	22.0 (21.4)
Glaucoma disease	3.5 (21/600)	14.9 (19.1)	9.4 (19.5)
No ocular pathology	19.5 (117/600)	28.9 (28.2)	22.2 (26.9)

Comment 16: In Table 3, for the row corresponding to diagnosis, how can there is one beta-coefficient? Since diagnosis is a categorical variable that include several ocular disease categories, there should be several beta-coefficients. Please provide the mean outer retinal band length for each ocular diagnosis.

Comment 17: Table 3: please clarify what beta-coefficient corresponds to. For example, for beta-coefficient for sex, what is the reference group, for beta-coefficient of age, does this coefficient correspond to per year increase in age? For beta coefficient of diagnostic category, what is the reference group?

Response (to comments 16 & 17): We agree that presenting diagnosis as a stand-alone variable is confusing with respect to beta coefficient interpretation. Table 3 has been modified such that univariate associations between specific disease categories and HORB length are nested under the 'diagnosis' subheading to allow recording of specific beta coefficients for each disease category. Data associated with the original 'diagnosis' variable has also been removed.

The reference group has also been added to all variables in Table 3 (see brackets in the first column). The variable 'high definition B-scan', originally based on A-scan density per B-scan for different OCT volumes, has also been removed due to high collinearity with the 'Device type' variable. The revised results show that HORB estimations of non-foveal B-scans are significantly associated with OCT device type and actually independent of B-scan length. Table 3 and the discussion (see response to comment 18) have been updated to reflect these changes:

Changes highlighted in green:

Table 3: Association between patient/scan characteristics and HORB length

Univariable analysis	Foveal B-scans		Non-foveal B-scans	
	p	β^{\dagger}	p	β
Age (years)	<0.001	-0.21	<0.001	-0.33
Sex (reference: female)	0.64	-0.02	0.87	0.33
VA (logMAR)	<0.001	-0.20	0.001	-27.06
BVS (Dioptres)	0.006	0.11	0.14	0.51
B-scan length (mm)	0.08	0.07	0.31	-0.04
Eccentricity (mm from fovea)	-	-	0.13	-0.06
Device type [‡] (reference: Spectralis OCT)	0.40	0.04	0.01	0.11
Diagnosis (reference: absence)				
Glaucoma disease	0.03	-14.0	0.06	-0.08
Glaucoma suspect	0.81	0.83	0.38	-2.71
Macular disease	0.06	-7.00	0.06	-6.25
Retinal disease	0.40	-3.45	0.61	-1.88
Optic nerve disease	0.83	1.27	0.97	-0.23
Outer retinal disease	0.06	-4.61	0.24	-2.63
Multiple linear regression analysis[§]				
Age (years)	<0.0001	-0.22	<0.0001	-0.24
VA (logMAR)	0.001	-0.13	0.09	-0.07
BVS (Dioptres)	<0.001	0.16	<0.001	0.12
B-scan length (mm)	0.07	0.07	-	-
Eccentricity (mm from fovea)	-	-	0.002	-0.13
Device type (reference: Spectralis OCT)	-	-	0.002	0.13
Diagnosis (reference: absence)				
Glaucoma disease	0.24	-0.05	0.25	-0.05
Macular disease	0.93	0.005	0.99	-0.001
Outer retinal disease	0.29	-0.06	0.87	-0.01

(Page 8-9, line 284)

Comment 18: The relationship of outer retinal band length with age, gender, visual acuity, refraction and eccentricity should be evaluated among eyes without any ocular disease. Current evaluation among all subjects with mixture of various ocular disease is complicated, and there may be many other confounding factors associated disease can distort the association.

Response: As suggested by the reviewer, a multiple regression analysis was performed for eyes without any ocular disease to determine whether the originally reported associations persist in this population (presented in **Table S3**). We found that HORB length was still associated with age for all images in this subgroup, but not visual acuity and refractive error for foveal B-scans and eccentricity and device type for non-foveal B-scans. We surmised that these differences may arise from variances in cataract density and proportions of pseudophakic patients between disease and non-disease groups. The results and discussion have been updated to reflect this finding and previous explanations for associations between refractive error, eccentricity and B-scan length and HORB length have been removed:

Results

A sub-analysis of patients without pathology was also conducted to assess for disease-related confounding factors. Associations were preserved for age for foveal and non-foveal B-scans. For foveal B-scans, associations to visual acuity ($p=0.09$) and refractive error ($p=0.089$) were no longer significant, as for eccentricity ($p=0.93$) and device type ($p=0.18$) for non-foveal B-scans in eyes without ocular disease, contrasting with the whole cohort analysis. (Page 8, lines 277-282)

Discussion

We also identified refractive error, eccentricity and device type as additional covariables affecting HORB length that have not been previously reported. However, in our subset analysis of eyes without

ocular disease, only age remained significantly associated with both foveal and non-foveal HORB length while refractive error significantly affected non-foveal HORB but not foveal HORB length. These differences could be explained by the effects of cataracts and pseudophakia on visual acuity and refraction, particularly as the diseased group was statistically older. The lack of association between B-scan eccentricity and device type to HORB length in eyes without ocular pathology likely stems from differences in clinical decision-making processes between groups as non-foveal B-scans are selected randomly in eyes without pathology and OCT devices may be chosen based on imaging capabilities. For example, peripheral OCT B-scans are more easily obtained using Spectralis OCT devices¹⁵ and thus are more likely to be used when peripheral pathology is identified. (Page 10, lines 324-335)

Reviewer 2: Dr. Xincheng Yao, University of Illinois at Chicago

Comment 19: This manuscript examined the outer retinal band appearance observed in OCT images using magnitude estimation. The subject involved for this study included an unbiased population. The reliability of measuring outer retinal band length using this method has been determined and its association with clinical outcomes has been assessed. The relevance of using such measurement for the examination of visual acuity and retinal disease diagnosis in an unselected patient population has been studied. This study concluded that the hyper outer reflective retinal band (HORB) length is reliably assessed using magnitude estimation and may be useful as a surrogate biomarker of visual acuity. Several factors affect HORB length estimations, which may contribute to the lack of association to retinal disease and highlights the need for covariable adjustment when examining HORB disruptions.

The rationale behind magnitude estimation as a reliable technique in OCT B-scans has not been established clearly. The presented results were obtained manually by only two graders. A thorough comparative assessment has not been conducted to validate the reliability of this approach. The method section needs more clarity with better visual depiction. Considering these issues, further improvements are required for publication.

Response: Thank you for providing a thorough evaluation of the manuscript. As suggested, additional information on magnitude estimation and the rationale behind the technique is detailed in the response to comment 20 and further clarity on the grading approach is provided in response to comment 21 (see below). Importantly, a third grader has been added to further validate the reliability of magnitude estimation for assessing outer retinal band length which produced results that are overall consistent with the reported intra and inter-rater reliability in the original manuscript (see comment 14).

Comment 20: The rationale behind using length of OCT B-scan as the reference and the length of the hyperreflective outer retinal bands (HORB) as the stimuli has not been discussed in detail. Other than pathological conditions, band disruptions in OCT can also occur due to noise artefacts, aberration, and even blood vessels. Hence the authors need to explain if this method is a well-established approach in OCT.

Response:

The reviewer raises an important point. Other causes of band disruptions besides retinal pathology were minimised by only including B-scans of meeting minimum quality control criteria (Cirrus B-scans: Signal strength > 6; Spectralis scans: Quality > 15), choosing graders experienced with OCT interpretation and providing instructions to graders that specifically advised them to ignore image artefacts including shadowing and noise. The methods have been updated to clarify this and the grading instructions have been included to the supplementary material:

Methods

All graders were clinicians experienced in OCT interpretation to ensure that gradings were not biased by image artefacts such as blood vessel shadowing and only images meeting minimum quality control criteria were included to control for noise. (Page 5, lines 176-179)

Supplementary

OCT B-scans were graded using the following instructions: 'Not counting the area directly under the optic nerve (if your image contains an optic nerve), what percentage of the horizontal length of the scan shows three complete and continuous hyperreflective lines in the retinal layers between the myoid zone and choriocapillaris? Ignore padding between around the OCT B-scan, shadowing and areas under the optic nerve. (Supplementary page 1, lines 4-9)

The application of magnitude estimation to assess outer retinal band length is novel, however magnitude estimation is a well-established psychophysical technique that underpins other ocular assessments already performed in eye care. The advantage of magnitude estimation is that it is a ratio measure which provides unitless assessments translatable across clinical contexts e.g. different B-scan length across OCT devices. The details of existing applications of magnitude estimation in eye care have been provided in the methods:

This approach was used as the magnitude estimation technique requires subjects to assess relative values (i.e. ratios) rather than actual length or size of stimuli in real units. Our approach is also consistent with other applications of magnitude estimation for evaluating ocular structures such as using optic disc size as the reference for cup-disc ratio estimations ratio¹⁷ or the width of a large vein at the disc margin to estimate drusen size for classification of age-related macular degeneration.¹⁸ (Page 5, lines 163-167)

3. Landa G, Su E, Garcia PM, Seiple WH, Rosen RB. Inner segment–outer segment junctional layer integrity and corresponding retinal sensitivity in dry and wet forms of age-related macular degeneration. *Retina*. 2011;31(2):364-70.
4. Fowler PV, Al-Ani AH, Thompson JM. Comparison of reliability of categorical and continuous scales for radiographic assessments of bone infill following secondary alveolar bone grafting. *The Cleft Palate-Craniofacial Journal*. 2018;55(2):269-75.
5. Masmali AM, Murphy PJ, Purslow C. Development of a new grading scale for tear ferning. *Contact Lens and Anterior Eye*. 2014;37(3):178-84.
6. Jain A, Saxena S, Khanna VK, Shukla RK, Meyer CH. Status of serum VEGF and ICAM-1 and its association with external limiting membrane and inner segment-outer segment junction disruption in type 2 diabetes mellitus. *Molecular vision*. 2013;19:1760.
7. Nakamura T, Ueda-Consolvo T, Oiwake T, Hayashi A. Correlation between outer retinal layer thickness and cone density in patients with resolved central serous chorioretinopathy. *Graefe's Archive for Clinical and Experimental Ophthalmology*. 2016;254(12):2347-54.
8. Wu Z, Ayton LN, Guymer RH, Luu CD. Second reflective band intensity in age-related macular degeneration. *Ophthalmology*. 2013;120(6):1307-8. e1.
9. Keenan TD, Clemons TE, Domalpally A, Elman MJ, Havilio M, Agrón E, et al. Retinal Specialist versus Artificial Intelligence Detection of Retinal Fluid from OCT: Age-Related Eye Disease Study 2: 10-Year Follow-On Study. *Ophthalmology*. 2021;128(1):100-9.
10. Scheetz J, Koklanis K, McGuinness M, Long M, Morris ME. Gaze behaviour and accuracy among novice and glaucoma specialist orthoptists during optic disc examination: A cross sectional study. *Australian Orthoptic Journal*. 2019;51:17-24 PubMed .
11. Cheung R, Ho S, Ly A. Optometrists' attitudes toward using OCT angiography lag behind other retinal imaging types. *Ophthalmic and Physiological Optics*. 2023.
12. Safi S, Thiessen T, Schmailzl KJ. Acceptance and resistance of new digital technologies in medicine: qualitative study. *JMIR research protocols*. 2018;7(12):e11072.
13. Giavarina D. Understanding bland altman analysis. *Biochemia medica*. 2015;25(2):141 PubMed - 51.
14. Koo TK, Li MY. A Guideline of Selecting and Reporting Intraclass Correlation Coefficients for Reliability Research. *J Chiropr Med*. 2016;15(2):155 PubMed -63. Epub 2016/06/23.

15. Bursac Z, Gauss CH, Williams DK, Hosmer DW. Purposeful selection of variables in logistic regression. Source code for biology and medicine. 2008;3(1):1-8.
16. Aumann S, Donner S, Fischer J, Müller F. Optical coherence tomography (OCT): principle and technical realization. High resolution imaging in microscopy and ophthalmology: new frontiers in biomedical optics. 2019:59-85.
17. Feuer WJ, Parrish II RK, Schiffman JC, Anderson DR, Budenz DL, Wells M-C, et al. The Ocular Hypertension Treatment Study: reproducibility of cup/disk ratio measurements over time at an optic disc reading center. American journal of ophthalmology. 2002;133(1):19-28.
18. Age-Related Eye Disease Study Research Group. A simplified severity scale for age-related macular degeneration: AREDS report no. 18. Archives of ophthalmology. 2005;123(11):1570.
19. Staurengi G, Sadda S, Chakravarthy U, Spaide RF. Proposed lexicon for anatomic landmarks in normal posterior segment spectral-domain optical coherence tomography: the IN• OCT consensus. Ophthalmology. 2014;121(8):1572 PubMed -8.

VERSION 2 – REVIEW

REVIEWER	Ying, Gui-shuang University of Pennsylvania Perelman School of Medicine, Ophthalmology
REVIEW RETURNED	22-Nov-2023
GENERAL COMMENTS	Thanks for addressing all my previous comments. The paper is improved.
REVIEWER	Yao, Xincheng University of Illinois at Chicago
REVIEW RETURNED	24-Nov-2023
GENERAL COMMENTS	My comments have been addressed.